# A metabolic profile of all-cause mortality risk identified in an observational study of 44,168 individuals

Joris Deelen [iD] et al.[#]

Predicting longer-term mortality risk requires collection of clinical data, which is often cumbersome. Therefore, we use a well-standardized metabolomics platform to identify metabolic predictors of long-term mortality in the circulation of 44,168 individuals (age at baseline 18–109), of whom 5512 died during follow-up. We apply a stepwise (forward-backward) procedure based on meta-analysis results and identify 14 circulating biomarkers independently associating with all-cause mortality. Overall, these associations are similar in men and women and across different age strata. We subsequently show that the prediction accuracy of 5- and 10-year mortality based on a model containing the identified biomarkers and sex (C-statistic = 0.837 and 0.830, respectively) is better than that of a model containing conventional risk factors for mortality (C-statistic = 0.772 and 0.790, respectively). The use of the identified metabolic profile as a predictor of mortality or surrogate endpoint in clinical studies needs further investigation.

Robust predictors of intermediate- and long-term mortality may be valuable instruments in clinical trials and medical decision-making. Predicting mortality in the final year of the life of a patient is generally feasible because of the abundance of available clinical data[1]. There is no consensus on the ultimate set of predictors of longer-term (5–10 years) mortality risk, since the predictive power of the currently used risk factors is limited[2], especially at higher ages. However, it is especially this age group and follow-up time window for which a robust tool would aid clinicians in assessing whether treatment is still sensible. Some of the currently used risk factors for mortality, such as systolic blood pressure and total cholesterol, show opposite associations with mortality in the elderly (i.e., above 85 years) as compared to middle age[3,4]. This could be due to mortality crossover of these risk factors or metabolic shifts that are difficult to predict in individuals[5,6], thus making them less suitable for accurate prediction of mortality in older individuals. Given the multimorbidity among older people, predictors of intermediate- and long-term mortality should ideally represent generic immune-metabolic health adversity rather than only being indicators of specific pathology. The number of molecular scores that are able to predict mortality across all ages is currently limited.

Fischer and colleagues used a high-throughput and well-standardized nuclear magnetic resonance (NMR) platform and identified four metabolic biomarkers, i.e., albumin, glycoprotein acetyls (GlycA), mean diameter for very low-density lipoprotein (VLDL) particles and citrate that are independently associated with all-cause and cause-specific (cardiovascular disease (CVD) and cancer) mortality[7–9]. The same metabolomics platform had also been utilized to predict CVD and type 2 diabetes[10–12]. Although the initial sample size of the study by Fischer and colleagues was large, the statistical power of the study was limited due to the relatively small number of observed deaths ($n = 684$) and under-representation of older individuals.

The current metabolomics study is the largest thus far, and includes 44,168 individuals (from 12 cohorts), spanning a wide age range. We first determine which metabolic biomarkers independently associate with prospective mortality in all individuals. Subsequently, we test the association of the biomarkers with mortality in different age strata. In the FINRISK 1997 cohort, consisting of 7603 individuals of whom 1213 died during follow-up, we compare the predictive value of a score based on the identified mortality-associated biomarkers with a score based on conventional risk factors for mortality.

## Results

**Association of metabolic biomarkers with all-cause mortality.** The primary survival meta-analysis of the 226 metabolic biomarkers for all-cause mortality was performed in 44,168 individuals from 12 cohorts, of whom 5512 died during follow-up (mean follow-up time per study ranging from 2.76 to 16.70 years) (Table 1). As depicted in Supplementary Data 1, 136 of the biomarkers showed a significant association with all-cause mortality after adjustment for multiple testing. When we subsequently adjusted for the 4 previously identified metabolic biomarkers[9], i.e., albumin, GlycA, VLDL particle size and citrate, the number of significant biomarkers increased to 159 (including the 4 previously identified biomarkers) (Supplementary Data 1). As the majority of the associated biomarkers are highly correlated, we tried to identify all independent biomarkers that were significantly associated with mortality. For this, we used a stepwise (forward-backward) procedure. To decrease the chance of overfitting, we only included a subset of 63 biomarkers in this step (Supplementary Data 2). Since the associations of the biomarkers with mortality in the primary survival analysis were similar in men and women (Supplementary Data 3), we performed the secondary analyses in men and women combined to increase power. After the stepwise procedure, 14 biomarkers showed to be independently associated with mortality. For the total lipids in chylomicrons and extremely large VLDL and small high-density lipoprotein (HDL), the mean diameter for VLDL particles, the ratio of polyunsaturated fatty acids to total fatty acids, and the concentrations of histidine, leucine, valine, and albumin a higher level is associated with decreased mortality, while for the concentrations of glucose, lactate, isoleucine, phenylalanine, acetoacetate, and GlycA the opposite applies (Table 2 and Supplementary Data 1). Of note, of the 4 previously identified mortality-associated biomarkers, only citrate was not selected in the fully adjusted model with 14 biomarkers due to its limited additional contribution. An increase of one unit in the metabolic biomarker score based on the 14 identified biomarkers, which ranges between −2 and 3 in most cohorts (see Supplementary Fig. 1, for examples), is associated with a 2.73 times higher mortality risk (HR = 2.73, 95% CI: 2.60–2.86, $P < 1.00 \times 10^{-132}$). The forest plots for each of these 14 biomarkers, based on the fully adjusted model, and the biomarker score are depicted in Supplementary Figs. 2–16.

**Association of biomarkers with disease-specific mortality.** To determine whether the identified biomarkers are indicators of disease-specific mortality risk, we also explored the associations of the biomarkers with cardiovascular, cancer, and infection-related mortality in the FINRISK 1997 cohort. As indicated in Table 3, the majority of the biomarkers associated with multiple mortality outcomes in the same direction as observed for all-cause mortality, including nonlocalized infections, thus representing general markers of health and disease, although some biomarkers, such as

### Table 1 Description of the cohorts included in this study

| Study | N | Males (%) | Deaths | Age at inclusion (range) | Mean follow-up time (SD) |
|---|---|---|---|---|---|
| Alpha Omega Cohort | 568 | 428 (75.4%) | 157 | 69.21 (59.31–80.94) | 7.79 (2.44) |
| ALSPAC | 4351 | 0 (0%) | 17 | 47 (34–63) | 5.69 (0.01) |
| EGCUT | 10,988 | 4106 (37.4%) | 912 | 46.10 (18–103) | 7.97 (1.77) |
| ERF study | 680 | 307 (45.1%) | 107 | 50.44 (18.10–86.50) | 10.67 (2.22) |
| FINRISK 1997 cohort | 7603 | 3778 (49.7%) | 1213 | 48.29 (24.15–74.28) | 16.70 (3.23) |
| DILGOM study | 4816 | 2256 (46.8%) | 190 | 52.39 (24.06–74.18) | 7.73 (0.75) |
| KORA F4 | 1790 | 871 (48.7%) | 123 | 60.89 (32–81) | 8.02 (1.25) |
| LLS nonagenarians | 843 | 326 (38.7%) | 823 | 97.35 (89.13–109.85) | 4.03 (3.09) |
| LLS offspring + partners | 2241 | 999 (44.6%) | 191 | 70.93 (42.54–91.25) | 11.76 (1.99) |
| PROSPER | 5329 | 2583 (48.5%) | 467 | 75.30 (69.37–83.39) | 2.76 (0.53) |
| Rotterdam Study | 2963 | 1241 (41.9%) | 1254 | 75.00 (52.21–98.13) | 8.28 (3.18) |
| TwinsUK | 1996 | 0 (0%) | 58 | 64.58 (42.37–87.84) | 4.32 (2.47) |

*SD* standard deviation

**Table 2 Association of the 14 identified metabolic biomarkers with all-cause mortality in the fully adjusted model**

| Biomarker | Full name | HR | 95% CI | P | $I^2$ | P (het) |
|---|---|---|---|---|---|---|
| XXL-VLDL-L | Total lipids in chylomicrons and extremely large VLDL | 0.80 | 0.75–0.85 | $1.53 \times 10^{-13}$ | 0.08 | 0.363 |
| S-HDL-L | Total lipids in small HDL | 0.87 | 0.84–0.90 | $5.98 \times 10^{-19}$ | 0.52 | 0.018 |
| VLDL-D | Mean diameter for VLDL particles | 0.85 | 0.80–0.90 | $8.51 \times 10^{-8}$ | 0.21 | 0.241 |
| PUFA/FA | Ratio of polyunsaturated fatty acids to total fatty acids (%) | 0.78 | 0.75–0.80 | $1.06 \times 10^{-47}$ | 0.71 | $8.65 \times 10^{-5}$ |
| Glc | Glucose | 1.16 | 1.13–1.19 | $2.22 \times 10^{-29}$ | 0.56 | 0.008 |
| Lac | Lactate | 1.06 | 1.03–1.10 | $6.24 \times 10^{-5}$ | 0.28 | 0.173 |
| His | Histidine | 0.93 | 0.90–0.96 | $1.15 \times 10^{-5}$ | 0.24 | 0.213 |
| Ile | Isoleucine | 1.23 | 1.14–1.32 | $2.14 \times 10^{-8}$ | 0.39 | 0.078 |
| Leu | Leucine | 0.82 | 0.76–0.89 | $7.34 \times 10^{-7}$ | 0.35 | 0.109 |
| Val | Valine | 0.87 | 0.82–0.92 | $1.04 \times 10^{-6}$ | 0.07 | 0.376 |
| Phe | Phenylalanine | 1.13 | 1.09–1.17 | $2.39 \times 10^{-12}$ | 0.44 | 0.052 |
| AcAce | Acetoacetate | 1.08 | 1.05–1.11 | $1.73 \times 10^{-8}$ | 0.35 | 0.108 |
| Alb | Albumin | 0.89 | 0.87–0.92 | $9.96 \times 10^{-13}$ | 0.52 | 0.017 |
| GlycA | Glycoprotein acetyls | 1.32 | 1.27–1.38 | $7.45 \times 10^{-41}$ | 0.45 | 0.046 |

*HR* hazard ratio, *CI* conference interval, *P* P value, $I^2$ heterogeneity statistic, *het* heterogeneity, *VLDL* very low-density lipoprotein particle, *HDL* high-density lipoprotein. The statistics in this Table have been generated with the R-package meta using the survival analyses results from the individual cohorts as input.

**Table 3 Association of the 14 identified metabolic biomarkers with all-cause and cause-specific mortality in the FINRISK 1997 cohort**

| Biomarker | N | All-cause (HR, 95% CI, P) | Cancer (HR, 95% CI, P) | Cardiovascular (HR, 95% CI, P) | Nonlocalized infections (HR, 95% CI, P) | Other (HR, 95% CI, P) |
|---|---|---|---|---|---|---|
| XXL-VLDL-L | 7583 | 0.77, 0.68–0.86, $1.00 \times 10^{-5}$ | 0.78, 0.64–0.95, 0.016 | 0.85, 0.73–0.99, 0.039 | 0.47, 0.26–0.86, 0.014 | 0.57, 0.40–0.82, 0.002 |
| S-HDL-L | 7583 | 0.95, 0.89–1.01, 0.085 | 0.89, 0.80–0.98, 0.023 | 0.96, 0.88–1.04, 0.319 | 0.80, 0.60–1.07, 0.130 | 1.00, 0.85–1.19, 0.966 |
| VLDL-D | 7583 | 0.99, 0.88–1.11, 0.802 | 0.95, 0.78–1.16, 0.619 | 1.06, 0.91–1.24, 0.462 | 1.89, 1.01–3.54, 0.045 | 0.90, 0.66–1.23, 0.520 |
| PUFA/FA | 7583 | 0.73, 0.69–0.78, $<2.22 \times 10^{-16}$ | 0.74, 0.66–0.83, $2.31 \times 10^{-7}$ | 0.77, 0.70–0.84, $2.77 \times 10^{-9}$ | 0.50, 0.37–0.68, $9.01 \times 10^{-6}$ | 0.69, 0.58–0.82, $4.06 \times 10^{-5}$ |
| Glc | 7583 | 1.13, 1.09–1.18, $5.85 \times 10^{-9}$ | 1.05, 0.96–1.13, 0.273 | 1.16, 1.10–1.22, $1.29 \times 10^{-8}$ | 1.05, 0.85–1.29, 0.643 | 1.03, 0.88–1.20, 0.721 |
| Lac | 7583 | 1.07, 1.00–1.14, 0.038 | 1.02, 0.92–1.14, 0.694 | 1.14, 1.05–1.23, 0.003 | 1.08, 0.77–1.52, 0.656 | 0.97, 0.83–1.14, 0.727 |
| His | 7583 | 0.93, 0.86–0.99, 0.031 | 0.88, 0.78–0.98, 0.023 | 0.89, 0.81–0.97, 0.010 | 1.12, 0.78–1.62, 0.538 | 1.11, 0.93–1.31, 0.250 |
| Ile | 7583 | 1.12, 0.95–1.32, 0.180 | 1.07, 0.81–1.41, 0.638 | 1.05, 0.84–1.32, 0.661 | 0.57, 0.24–1.38, 0.214 | 1.26, 0.84–1.88, 0.270 |
| Leu | 7583 | 0.80, 0.67–0.97, 0.020 | 0.89, 0.65–1.22, 0.474 | 0.73, 0.57–0.94, 0.016 | 1.02, 0.40–2.62, 0.967 | 0.83, 0.53–1.30, 0.419 |
| Val | 7583 | 0.89, 0.79–1.02, 0.084 | 0.95, 0.77–1.19, 0.672 | 1.05, 0.88–1.26, 0.580 | 1.32, 0.67–2.61, 0.425 | 0.64, 0.48–0.86, 0.003 |
| Phe | 7583 | 1.21, 1.10–1.33, $4.17 \times 10^{-5}$ | 1.14, 0.98–1.33, 0.087 | 1.28, 1.13–1.44, $7.61 \times 10^{-5}$ | 1.21, 0.75–1.96, 0.437 | 1.07, 0.84–1.36, 0.575 |
| AcAce | 7583 | 1.06, 1.00–1.13, 0.038 | 1.04, 0.94–1.15, 0.441 | 1.07, 0.99–1.15, 0.096 | 1.23, 0.90–1.69, 0.191 | 1.11, 0.96–1.28, 0.167 |
| Alb | 7583 | 0.89, 0.83–0.96, 0.003 | 0.90, 0.79–1.01, 0.084 | 0.88, 0.79–0.97, 0.009 | 0.81, 0.55–1.17, 0.261 | 1.00, 0.83–1.21, 0.997 |
| GlycA | 7583 | 1.41, 1.26–1.57, $5.75 \times 10^{-10}$ | 1.35, 1.12–1.61, 0.001 | 1.36, 1.18–1.57, $3.13 \times 10^{-5}$ | 2.03, 1.15–3.59, 0.015 | 1.52, 1.16–2.01, 0.003 |

The number of (cause-specific) deaths were 1210 (all-cause), 434 (cancer), 687 (cardiovascular), 43 (nonlocalized infections), and 189 (other). *N* number of samples, *HR* hazard ratio, *CI* conference interval, *P* P value. The statistics in this Table have been generated with the R-package survival

glucose, seem to be risk factors for a specific mortality outcome, in this case cardiovascular-related mortality.

**Association of metabolic biomarkers across the lifespan**. To investigate the association across the lifespan for the mortality-associated biomarkers identified in this study, we performed age-stratified mortality analyses. All 14 biomarkers that were part of the fully adjusted model showed consistent associations with mortality across all strata (Supplementary Data 4) and the same was true for the metabolic biomarker score (Supplementary Fig. 17).

**Mortality risk prediction accuracy of identified biomarkers**. To determine the mortality risk prediction accuracy of the 14 identified biomarkers, we generated weighted risk scores based on conventional risk factors and on our identified biomarkers plus sex. The weights of the risk scores were estimated in the Estonian Biobank cohort and the FINRISK 1997 cohort was used as validation cohort to compare the mortality risk prediction accuracy of the models. Instead of looking at the added value of the individual biomarkers, we directly compared the two models to determine if this single point NMR measurement on itself could be used as a standard for risk assessment of mortality. Removal of FINRISK 1997 from the discovery analysis resulted in similar

effect estimates as those reported in Table 2, indicating that it is unlikely that the risk prediction analyses are influenced by overfitting. Given the restricted follow-up time in the elderly cohorts and the need for mortality risk indicators in the clinic at higher ages, we investigated both 5- and 10-year mortality in all individuals as well as only in those above 60 years of age. As depicted in Fig. 1 and Table 4, the *C*-statistic was 0.065 ($P = 5.48 \times 10^{-4}$) or 0.040 ($P = 2.48 \times 10^{-5}$) larger when comparing the model with the 14 biomarkers (*C*-statistic = 0.837 and 0.830) to the model with conventional risk factors (*C*-statistic = 0.772 and 0.790) when looking at 5- or 10-year mortality, respectively. The difference in the *C*-statistic was even larger when only individuals above 60 years of age were included (Table 4). Reclassification analyses showed higher integrated discrimination improvement (IDI) (e.g., + 8.6% ($P = 1.83 \times 10^{-12}$) for 10-year mortality) when comparing the model with the biomarkers to the model with conventional risk factors (Table 4). When compared to a model with the 4 previously identified mortality-associated biomarkers, the model with the 14 biomarkers also showed higher *C*-statistics and IDI's for both 5- and 10-year mortality (Supplementary Table 1). Since the conventional risk factors were only partially correlated with the 14 biomarkers (Supplementary Fig. 18), we also compared a model with the biomarkers to a model combining the conventional risk factors and biomarkers to

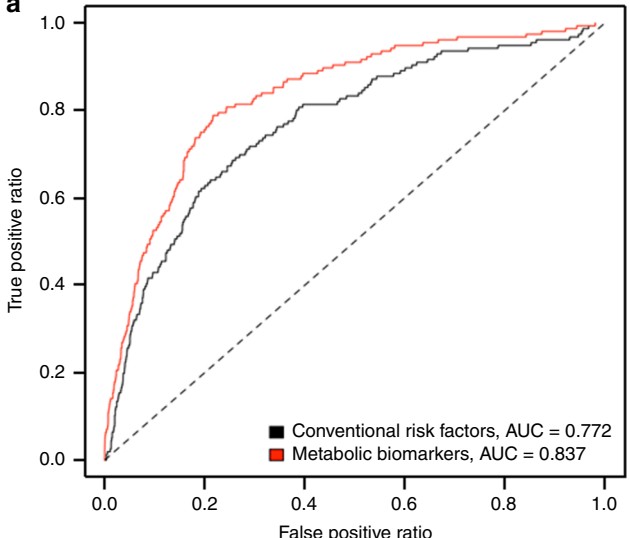

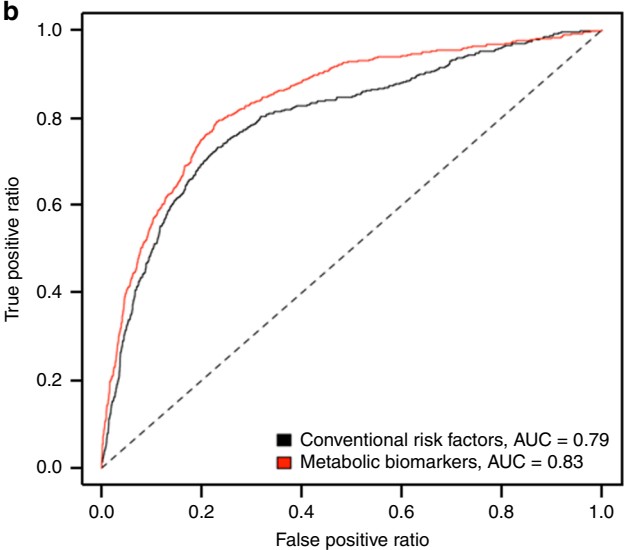

**Fig. 1** Mortality risk prediction accuracy of the 14 identified metabolic biomarkers. Receiver operating characteristic curves for 5- (**a**) and 10-year (**b**) mortality in the FINRISK 1997 cohort. The curves are based on the predictions from the conventional risk factors (black) and the metabolic biomarkers (red). AUC area under the curve

see if the addition of conventional risk factors could further improve the mortality risk prediction accuracy. However, this was not the case (Supplementary Table 2).

**Reproducibility and validation of metabolic biomarkers**. The reproducibility of the quantification of the 14 identified biomarkers, which was determined using previously generated in-house NMR from the Leiden Longevity Study (LLS) offspring + partners and nonagenarians[13,14], was very good (all $r > 0.8$, Supplementary Fig. 19). This, in combination with the previously published validation of some additional identified biomarkers with other techniques (i.e., the ratio of polyunsaturated fatty acids to total fatty acids and the concentrations of albumin and GlycA)[10,15,16], and the provided data on the consistency of the identified mortality-related small molecules (i.e., the concentrations of glucose, lactate, histidine, isoleucine, leucine, valine, and phenylalanine) as measured with other widely used metabolomics platforms (i.e., Metabolon and Biocrates, Supplementary Table 3)

show evidence of high analytical consistency with other biomarker assays, providing confidence that our findings should be reproducible when the metabolic biomarkers would be measured using other metabolomics platforms or techniques.

## Discussion

By performing high-throughput metabolic biomarker profiling in 44,168 individuals from 12 cohorts, we identified a set of 14 biomarkers independently associating with all-cause mortality. The associations of these biomarkers were consistent in men and women and across age strata. The identified biomarkers represent general health up to the highest ages rather than specific disease-related death causes. In combination, these biomarkers clearly improve risk prediction of 5- and 10-year mortality as compared to conventional risk factors across all ages. These results suggest that metabolic biomarker profiling could potentially be used to guide patient care, if further validated in relevant clinical settings.

Our results show that the use of an affordable, well-standardized, and high-throughput NMR platform measuring multiple biomarkers leads to a high mortality risk prediction accuracy. We observed similar effects of the biomarkers on mortality in the cohorts using either EDTA plasma (Alpha Omega Cohort, ERF study, FINRISK 1997 cohort, DILGOM study, LLS non-agenarians, LLS offspring + partners, PROSPER, and Rotterdam Study) or serum (ALSPAC, EGCUT, KORA F4, and TwinsUK). In addition, the associations of the identified biomarkers with mortality are independent of the sex, age and cause of death of the individuals, and are thus unaffected by mortality crossover. Hence, in comparison to conventional risk factors, such as systolic blood pressure and total cholesterol, these biomarkers seem much more suitable for guided screening of older individuals at risk, as surrogate endpoint in clinical trials among older individuals, and for targeted prevention of mortality.

The 14 identified biomarkers are involved in various processes, such as lipoprotein and fatty acid metabolism, glycolysis, fluid balance, and inflammation. Although the majority of these biomarkers have been associated with mortality before, this is the first study that shows their independent effect when combined into one model. In comparison to the previous study by Fischer et al.[9], we increased the sample size and number of deaths by fivefold and almost tenfold, respectively. This resulted in identification of more biomarkers (14 versus 4) and improved prediction accuracy. We were able to replicate the associations of all four biomarkers identified in the previous work. However, citrate was not included in our fully adjusted model, since this biomarker did not pass the multiple testing threshold. A possible explanation for this could be that one, or multiple, of the currently included biomarkers partially capture the effect of citrate, resulting in the attenuation of the association.

The total lipids in chylomicrons and extremely large VLDL and small HDL and the mean diameter for VLDL particles play a role in lipid metabolism and their association with mortality is likely caused by their involvement in the regulation of plasma triglyceride levels, a known risk factor for mortality[17]. The association of polyunsaturated fatty acids with different mortality outcomes has been attributed to its variety of actions, including its anti-inflammatory properties and inhibition of atherosclerosis[18]. The association between postprandial glucose levels and mortality is likely attributable to a loss in glycemic control[19], while the association of both albumin and GlycA with mortality has been attributed to their role in inflammation[16,20]. The association between the other identified biomarkers and mortality is less well described, although they all play a well-known role in health and disease[21–23]. Future studies should be performed to determine which health conditions are further reflected by the identified

**Table 4 Results of the discrimination and reclassification analyses for all-cause mortality in the FINRISK 1997 cohort comparing the conventional risk factor score with the metabolic biomarker score**

| Follow-up time (years) | Age | Conventional risk factor score *C*-statistic | Metabolic biomarker score *C*-statistic | Difference in *C*-statistic | IDI |
|---|---|---|---|---|---|
| 5 | All | 0.772 | 0.837 | $0.065 \pm 0.019$, $P = 5.48 \times 10^{-4}$ | $5.9 \pm 1.9\%$, $P = 0.001$ |
| 5 | >60 | 0.626 | 0.732 | $0.105 \pm 0.027$, $P = 0.0001$ | $8.6 \pm 2.1\%$, $P = 3.20 \times 10^{-5}$ |
| 10 | All | 0.790 | 0.830 | $0.040 \pm 0.010$, $P = 2.48 \times 10^{-5}$ | $8.6 \pm 1.2\%$, $P = 1.83 \times 10^{-12}$ |
| 10 | >60 | 0.650 | 0.715 | $0.065 \pm 0.014$, $P = 3.29 \times 10^{-6}$ | $11.9 \pm 1.5\%$, $P = 1.13 \times 10^{-14}$ |

The estimates for the risk scores were derived from the Estonian Biobank cohort. The conventional risk factor score, included sex, body mass index, systolic blood pressure, total cholesterol, high-density lipoprotein (HDL) cholesterol, triglycerides, creatinine, smoking status, alcohol consumption, and prevalent diabetes, cardiovascular disease, and cancer. The metabolic biomarker score, included total lipids in extremely large very low-density lipoprotein particle (VLDL), total lipids in small HDL, VLDL diameter, ratio of polyunsaturated fatty acids to total fatty acids, glucose, lactate, histidine, isoleucine, leucine, valine, phenylalanine, acetoacetate, albumin, glycoprotein acetyls, and sex. *IDI* integrated discrimination improvement. The statistics in this Table have been generated with custom-made functions in R

metabolic biomarkers and by what mechanisms. Such research is exemplified by previous work on the relation between metabolic biomarkers and all-cause mortality[9], which opened up new avenues for research into the metabolic biomarker GlycA[16].

For two of the biomarkers, i.e., the total lipids in extremely large VLDL lipids and isoleucine, the direction of effect changes when adjusting for the remaining 12 biomarkers. This change is most likely due to the inclusion of GlycA and the two other branch-chain amino acids, i.e., leucine and valine, in the model. Adjusting for GlycA removes the correlated negative effect of the total lipids in extremely large VLDL lipids, while adjusting for leucine and valine removes the correlated positive effect of isoleucine, resulting in appearance of opposite associations for these biomarkers. A similar effect was observed by Fischer et al.[9] for VLDL diameter after inclusion of GlycA in their model. It would be interesting to see if a similar effect is also observed for other phenotypes using multivariate adjusted models.

A potential limitation of our study is that the number of biomarkers captured by our targeted NMR platform is only a fraction of the metabolites in the human serum[24]. More complete high-throughput metabolic biomarker platforms are available, but these are usually more expensive. The predictive accuracy of these more complete platforms may be compared to the one used in this study. Efforts to increase the number of identifiable biomarkers using inexpensive high-throughput metabolic biomarker platforms (e.g., NMR or liquid chromatography–mass spectrometry) will likely result in identification of many more mortality-associated biomarkers and, hence, improved risk prediction.

Although we were able to show a good predictive ability of our biomarkers for mortality risk using two complementary methods (the *C*-statistic and IDI), the metabolic biomarker score constructed is not yet suitable for classification of patients in the clinic, since it is based on scaled biomarker values created separately for each cohort. Future efforts should therefore be focused on creation of a metabolic biomarker score that could be used for clinical research based on concentration units that could be generated using individual-level data.

In conclusion, we identified a set of 14 metabolic biomarkers that independently associate with all-cause mortality. A score based on these 14 biomarkers and sex leads to improved risk prediction as compared to a score based on conventional risk factors. This indicates that this affordable, well-standardized, and high-throughput NMR measurement may be used to generate a standard for risk assessment of mortality in the clinic. Such a score could potentially be used in clinical practice to guide treatment strategies, for example when deciding whether an elderly person is too fragile for an invasive operation. In addition, it may be used as a surrogate endpoint for clinical trials in older individuals, since showing (a reduction in) the total mortality endpoint is mostly not feasible due to the limited duration and number of cases in a regular clinical trial. The currently used

metabolomics platform can be incorporated in ongoing clinical studies to explore its value, opening up new avenues for research to establish the utility of metabolic biomarkers in clinical settings.

The summary statistics of our primary survival meta-analysis have been made publically available in the BBMRI -omics atlas: http://bbmri.researchlumc.nl/atlas.

## Methods

**Study populations**. The individuals included in this study were selected from the following cohorts; Alpha Omega Cohort, Avon Longitudinal Study of Parents and Children (ALSPAC), Estonian Biobank cohort, Erasmus Rucphen Family (ERF) study, FINRISK 1997 cohort, Dietary, Lifestyle and Genetic Predictors of Obesity and Metabolic Syndrome (DILGOM) study, Cooperative Health Research in the Region of Augsburg (KORA F4) study, LLS, PROspective Study of Pravastatin in the Elderly at Risk (PROSPER), Rotterdam Study (RS), and TwinsUK. All individuals were of European descent. A description of the cohorts is provided in Table 1 and the Supplementary Methods.

We have complied with all relevant ethical regulations for work with human subjects. All participants provided written informed consent, and the studies were approved by the relevant institutional review boards.

**Measurement of metabolic biomarkers**. The metabolic biomarkers were quantified from EDTA plasma and serum samples using high-throughput NMR metabolomics (Nightingale Health Ltd., Helsinki, Finland). This method provides simultaneous quantification of routine lipids, lipoprotein subclass profiling with lipid concentrations within 14 subclasses, fatty acid composition, and various low-molecular metabolites, including amino acids, ketone bodies, and gluconeogenesis-related metabolites, in molar concentration units. Details of the experimentation and applications of the NMR metabolomics platform have been described previously[8,25]. Several of the metabolic biomarkers have already been validated with other techniques (i.e., routine clinical chemistry assays, gas chromatography, an enzymatic method, and/or mass spectrometry)[8,10,15,16,26]. Furthermore, the metabolic biomarkers measured using the Nightingale Health platform have been used in numerous published epidemiological studies (see https://nightingalehealth.com/publications for an overview). The genetic work based on the Nightingale Health platform also underscores that the labels given to the metabolic biomarkers are correct and are associated with biologically relevant and plausible genes[27–29]. For the analyses in this study we first used all 226 available measurements, including the highly correlated lipid subclasses and compositions (for a full list see Supplementary Data 2). Due to the high correlation among the measurements, the selection of independently associated biomarkers was based on a subset of 63 biomarkers to prevent overfitting. The selection of these biomarkers was based on previous studies using this platform and the list comprises the total lipid concentrations, fatty acid composition, and low-molecular-weight metabolites, including amino acids, glycolysis-related metabolites, ketone bodies and metabolites involved in fluid balance and immunity (Supplementary Data 2)[10,12].

**Statistical analyses**. For each study, a value of one was added to all biomarkers containing zeroes (i.e., $x + 1$), which indicates the value was below the detection limit. Subsequently, all biomarkers were log-transformed and scaled to standard deviation units, separately per study. Similar to the previous study by Fischer et al.,[9] a Cox proportional hazards model with age at blood sampling as the time scale, was used to determine the associations of the biomarkers with all-cause mortality. In addition, the basic models were adjusted for age at blood sampling, sex and study-specific covariates that are related to demography and relatedness of the included individuals. Age at blood sampling was included in the model to make the results directly comparable with the age-stratified analyses (see below) in which the follow-up time of some individuals encompassed multiple age groups, so the age at sampling could have been before the age at start of the age group. To check for differences between sexes, we also performed sex-stratified analyses.

For the secondary analyses, we additionally adjusted the basic model for the 4 metabolic biomarkers previously reported by Fischer et al.,[9] i.e., albumin, GlycA, the mean diameter for VLDL particles and citrate (step 1), as well as 11 of the independently associating biomarkers discovered in the current study, i.e., the total lipids in chylomicrons and extremely large VLDL and small HDL, the ratio of polyunsaturated fatty acids to total fatty acids, and glucose, lactate, histidine, isoleucine, leucine, valine, phenylalanine, and acetoacetate levels (step 2). To select the biomarkers used for adjustment in step 2, we performed a stepwise (forward-backward) procedure based on successive rounds of meta-analyses. In each round we added to the model the unselected biomarker that showed the lowest $P$ value in the previous round of the stepwise procedure (forward step). Next, we removed biomarkers from the model if the previous step resulted in an increase of the $P$ value above the threshold (backward step). We stopped the procedure once all unselected biomarkers showed a $P$ value above the threshold in the working model. As threshold we used the Bonferroni-adjusted $P$ value to adjust for multiple testing (see below). To test the combined effect of the 14 identified biomarkers, we also created a metabolic biomarker score. To this end, the log-transformed and scaled biomarkers were multiplied by their weights, based on the meta-analyses results (i.e., ln(hazard ratio) from Table 2), and subsequently summed.

For the age-stratified analyses, samples were divided into age groups of <60, 60–70, 70–80, 80–90, and >90 years. Some samples were used multiple times, since their follow-up time encompassed two, or even three, age groups.

To determine the predictive value of the identified mortality-associated biomarkers, we constructed four weighted risk scores. The weights for the risk scores were based on the Estonian Biobank cohort (Supplementary Table 4) to avoid overestimation. The selection of the Estonian Biobank as training set was based on the fact that this was the largest dataset in our study containing data on most conventional risk factors for mortality, with the exception of C-reactive protein. The selection of conventional risk factors was based on the previous study by Fischer et al.[9] using the same dataset. The first risk score contains the conventional risk factors (i.e., sex, body mass index, systolic blood pressure, total cholesterol, HDL cholesterol, triglycerides, creatinine, smoking, alcohol, prevalent diabetes, prevalent CVD, and prevalent cancer). The second risk score contains our 14 identified independent mortality-associated biomarkers plus sex. The third risk score contains the four previously identified mortality-associated biomarkers plus sex. The fourth score contains our 14 identified independent mortality-associated biomarkers and the conventional risk factors (excluding total cholesterol, HDL cholesterol, triglycerides, and creatinine, since they were also part of the Nightingale Health platform). Age at sampling was not included in the risk scores, since this was used as the time scale. The predictive ability of the weighted risk scores was tested in the FINRISK 1997 cohort. We used two measures to assess the predictive value of the risk scores: (1) C-statistics and (2) IDI[30,31].

Biomarkers were considered significant when the $P$ value was below the Bonferroni-adjusted threshold of $2.21 \times 10^{-4}$ (0.05/226), which takes into account that we tested 226 biomarkers. The $P$ values for the difference between sexes and age strata were calculated using meta-analyses heterogeneity statistics ($I^2$). The survival analyses in the individual cohorts were performed using R and STATA/SE 11.2 (StataCorp LP, College Station, TX, USA), while the meta-analyses were performed using a fixed-effect model implemented in the R-package meta. The discrimination and reclassification analyses were performed using custom-made functions in R.

**Reporting summary**. Further information on research design is available in the Nature Research Reporting Summary linked to this article.

## Data availability
The datasets generated and/or analyzed during the current study are available from the corresponding author upon request. In addition, the quantified metabolic biomarker datasets of the cohorts that participated in this study are available upon request through http://www.bbmri.nl/omics-metabolomics/ (Alpha Omega Cohort, ERF study, LLS nonagenarians, LLS offspring + partners, and Rotterdam Study), http://www.bristol.ac.uk/alspac/researchers/access/ (ALSPAC), https://www.geenivaramu.ee/en/biobank.ee/data-access (EGCUT), https://thl.fi/en/web/thl-biobank/for-researchers/apply (FINRISK 1997 cohort and DILGOM study), https://epi.helmholtz-muenchen.de/ (KORA F4), https://twinsuk.ac.uk/resources-for-researchers/access-our-data/ (TwinsUK), and the PROSPER executive committee (J. Wouter Jukema; J.W.Jukema@lumc.nl). We are unable to share the raw NMR data from Nightingale Health Ltd., as the company holds the proprietary rights. The NMR data of the LLS samples that were used to test the reproducibility of the quantification of the identified metabolic biomarkers have been deposited in MetaboLights under accession code MTBLS974 (https://www.ebi.ac.uk/metabolights/MTBLS974)[32].

## Code availability
We have provided the most important scripts that we used for the scaling of the metabolic biomarkers, single cohort analyses (for the Alpha Omega Cohort, as example), and meta-analyses as Supplementary Datas 5–7. The custom-made R functions used to perform the discrimination and reclassification analyses are available from the corresponding author upon request.

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

## Acknowledgements

This work was performed within the framework of the BBMRI Metabolomics Consortium funded by BBMRI-NL, a research infrastructure financed by the Dutch government (NWO 184.021.007 and 184.033.111). A full list of acknowledgments, including support for each study, is provided in Supplementary Note 1.

## Author contributions

J.D. and P.E.S. designed the project, established and coordinated the consortium of studies, and wrote and revised the first and subsequent drafts of the paper. J.D., J.K., K.F., A.vdS., S.T., G.K., A.B., J.Z., A.V., C.M. and F.D. performed the data analyses. P.W. was responsible for the metabolic profiling at Nightingale Health Ltd. J.K., K.F., A.vdS., S.T., G.K., A.B., J.Z., E.B.vdA., M.A-K., N.A., A.D., M.G., D.vH., M.A.I., J.B.vK., S.P.M., A.P., V.S., N.S., T.D.S., H.T., A.V., M.W., P.W., G.D.S., A.M., M.P., C.M., J.M.G., F.D., M.B., J. W.J. and C.M.vD. contributed data and/or revised the draft critically for important intellectual content.

## Additional information

**Competing interests:** P.W. is an employee and shareholder of Nightingale Health Ltd., the company offering the NMR-based metabolite profiling used in the current study. J.K. owns stock options for Nightingale Health Ltd. V.S. has participated in a conference trip sponsored by Novo Nordisk and received a honorarium for participating in an advisory board meeting. He also has an ongoing research collaboration with Bayer Ltd. (all unrelated to the present study). Remaining authors declare no competing interests.

Joris Deelen [1,2], Johannes Kettunen [3,4], Krista Fischer[5], Ashley van der Spek[6], Stella Trompet[7,8], Gabi Kastenmüller[9,10,11], Andy Boyd [12], Jonas Zierer[9,11,13], Erik B. van den Akker [1,14], Mika Ala-Korpela [4,15,16,17,18,19], Najaf Amin[6], Ayse Demirkan[20,21], Mohsen Ghanbari [6,22], Diana van Heemst[7], M. Arfan Ikram [6,23,24], Jan Bert van Klinken [25,26], Simon P. Mooijaart[7], Annette Peters[10,27], Veikko Salomaa [3], Naveed Sattar [28], Tim D. Spector[11], Henning Tiemeier [6,29], Aswin Verhoeven[30], Melanie Waldenberger[27,31], Peter Würtz[32], George Davey Smith [17], Andres Metspalu [5,33], Markus Perola[34,35], Cristina Menni [11], Johanna M. Geleijnse [36], Fotios Drenos[17,37], Marian Beekman [1], J. Wouter Jukema [8], Cornelia M. van Duijn[6,38] & P. Eline Slagboom[1]

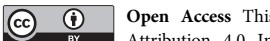

[1]Molecular Epidemiology, Department of Biomedical Data Sciences, Leiden University Medical Center, PO Box 9600, 2300 RC Leiden, The Netherlands. [2]Max Planck Institute for Biology of Ageing, PO Box 41 06 23, 50866 Cologne, Germany. [3]National Institute for Health and Welfare, PO Box 30, 00271 Helsinki, Finland. [4]Computational Medicine, Center for Life Course Health Research and Biocenter Oulu, University of Oulu, PO Box 5000, 90014 Oulu, Finland. [5]The Estonian Genome Center, University of Tartu, Riia 23b, 51010 Tartu, Estonia. [6]Department of Epidemiology, Erasmus Medical Center, PO Box 2040, 3000 CA Rotterdam, The Netherlands. [7]Department of Internal Medicine, section of Gerontology and Geriatrics, Leiden University Medical Center, PO Box 9600, 2300 RC Leiden, The Netherlands. [8]Department of Cardiology, Leiden University Medical Center, PO Box 9600, 2300 RC Leiden, The Netherlands. [9]Institute of Bioinformatics and Systems Biology, Helmholtz Zentrum München, Ingolstaedter Landstraße 1, 85764 Neuherberg, Germany. [10]German Center for Diabetes Research (DZD), Ingolstaedter Landstraße 1, 85764 Neuherberg, Germany. [11]Department of Twin Research and Genetic Epidemiology, King's College London, St Thomas' Hospital, Strand, London WC2R 2LS, UK. [12]ALSPAC, Population Health Sciences, Bristol Medical School, University of Bristol, Oakfield House, Oakfield Grove, Bristol BS8 2BN, UK. [13]Novartis Institutes for BioMedical Research, Novartis Campus, Fabrikstrasse 2, 4056 Basel, Switzerland. [14]The Delft Bioinformatics Lab, Delft University of Technology, PO Box 5031, 2600 GA Delft, The Netherlands. [15]Systems Epidemiology, Baker Heart and Diabetes Institute, PO Box 6492 Melbourne Victoria 3004, Australia. [16]Population Health Science, Bristol Medical School, University of Bristol, Oakfield House, Oakfield Grove, Bristol BS8 2BN, UK. [17]MRC Integrative Epidemiology Unit, Population Health Sciences, Bristol Medical School, University of Bristol, Oakfield House, Oakfield Grove, Bristol BS8 2BN, UK. [18]NMR Metabolomics Laboratory, School of Pharmacy, University of Eastern Finland, Yliopistonranta 1C, Kuopio 70210, Finland. [19]Department of Epidemiology and Preventive Medicine, School of Public Health and Preventive Medicine, Faculty of Medicine, Nursing and Health Sciences, The Alfred Hospital, Monash University, Melbourne, Victoria 3800, Australia. [20]Section of Statistical Multi-omics, Department of Clinical and Experimental research, University of Surrey, Guildford, Surrey GU2 7XH, UK. [21]Department of Genetics, University Medical Center Groningen, PO Box 30001, 9700 RB Groningen, The Netherlands. [22]Department of Genetics,

School of Medicine, Mashhad University of Medical Sciences, PO Box 91735-951, 9133913716 Mashhad, Iran. [23]Department of Radiology and Nuclear Medicine, Erasmus Medical Center, PO Box 2040, 3000 CA Rotterdam, The Netherlands. [24]Department of Neurology, Erasmus Medical Center, PO Box 2040, 3000 CA Rotterdam, The Netherlands. [25]Department of Human Genetics, Leiden University Medical Center, PO Box 9600, 2300 RC Leiden, The Netherlands. [26]Einthoven Laboratory for Experimental Vascular Medicine, Leiden University Medical Center, PO Box 9600, 2300 RC Leiden, The Netherlands. [27]Institute of Epidemiology II, Helmholtz Zentrum München, Ingolstaedter Landstraße 1, 85764 Neuherberg, Germany. [28]Institute of Cardiovascular and Medical Sciences, Cardiovascular Research Centre, University of Glasgow, 126 University Place, Glasgow G12 8TA, UK. [29]Department of Psychiatry, Erasmus University Medical Center-Sophia Children's Hospital, PO Box 2040, 3000 CA Rotterdam, The Netherlands. [30]Center for Proteomics and Metabolomics, Leiden University Medical Center, PO Box 9600, 2300 RC Leiden, The Netherlands. [31]Research Unit of Molecular Epidemiology, Helmholtz Zentrum München, Ingolstaedter Landstraße 1, 85764 Neuherberg, Germany. [32]Nightingale Health Ltd., Mannerheimintie 164a, 00300 Helsinki, Finland. [33]Institute of Molecular and Cell Biology, University of Tartu, Riia 23, 23b - 134, 51010 Tartu, Estonia. [34]Institute for Molecular Medicine Finland, University of Helsinki, Tukholmankatu 8, 00290 Helsinki, Finland. [35]Clinical and Molecular Metabolism Research Program, Faculty of Medicine, University of Helsinki, PO Box 63, 00014 Helsinki, Finland. [36]Division of Human Nutrition, Wageningen University, PO Box 17, 6700 AA Wageningen, The Netherlands. [37]Department of Life Sciences, Brunel University London, Uxbridge UB8 3PH, UK. [38]Leiden Academic Centre for Drug Research, Leiden University, PO box 9502, 2300 RA Leiden, The Netherlands

