## [Peer Review File · Nature Communications]

Reviewers' comments:

Reviewer #1 (Remarks to the Author):

The present study is the largest of its kind to correlate NMR-detectable plasma metabolic parameters with mortality. As such, the paper certainly deserves rapid publication in Nature Communications. Altogether, the methodology appears adequate and the data are convincing. Independent statistical review is recommended, however.

I only have some relatively minor suggestions.

In the Discussion the authors say "This resulted in [...] improved prediction accuracy (data not shown)." Since this is the crucial point of the manuscript, it would be good to demonstrate that prediction accuracy is indeed improved.

When reading the text without simultaneously analyzing Table 2 and 3, it is difficult to understand which among the metabolites or other analytes measured by NMR are positively or negatively influencing longevity. Some rewording might be helpful to improve the readability of the text.

Reviewer #2 (Remarks to the Author):

The manuscript by Deleen and colleagues details an impressive meta-analysis of 44,168 individuals using NMR spectroscopy to profile metabolic changes in blood plasma/serum and investigate their association with all-cause mortality. This extends previous studies by Wurtz et al. and Fisher et al. that had previously examined cardiovascular risk and all-cause mortality risk in a sub-set of the cohorts used in the present study using ¹H NMR spectroscopy. The major selling point of the present study is a large increase in mortality cases across the cohorts which significantly adds to the power of their statistics, and in particular has expanded the number of 'metabolites' associated with all-cause mortality compared with their previous publication by Fisher et al. However, while they have significantly increased the number of individuals in the present study I do wonder what this analysis adds to the previous studies. While they show the metabolic markers are more predictive than some classically used parameters we don't get to understand why these supposed metabolites are more predictive or what metabolic processes they represent. I feel this is a real missed opportunity and could be tackled by examining correlations between classic risk factors and the metabolic risk factors detected by NMR spectroscopy – for example how much are the effects of the different metabolites mediated through BMI? Which of the metabolites could be combined with the classical risk factors to achieve a comparable risk prediction to the 14 metabolites? This would help us understand which metabolites brought in new predictive power compared with the classic risk factors. I also wondered how much their classic risk factor score would be improved by including some simple clinical chemistry such as blood total triglyceride. I am also concerned about how many metabolites the authors really detect. There are not 226 independent metabolites detectable by NMR spectroscopy and in many cases this assay does not measure a metabolite as such but a chemical group found within a metabolite. However, the word metabolite is used even for quite confusing moieties such as the total lipid in chylomicrons or total lipids in extremely large VLDL. Also how can NMR work out a ratio of polyunsaturated fats to saturated fats when it can't predict what fatty acids are detected and hence the chain length of the lipids detected – certainly not what species are detected using only ¹H NMR spectroscopy in a complex mixture like blood. Presumably what they mean is that some of these resonances can be correlated with the concentrations of certain metabolites but there is a fundamental difference between say quantifying lactate where there are discrete resonances associated with only that metabolite or -CH₂- which could come from numerous fatty acids and in turn could be part of many different lipid species.

Data availability – there are a number of repositories for metabolomic data and the data should be made available in keeping with Nature guidelines concerning open access to data.

How can the association of metabolites with risk of all-cause death be independent of whether serum or EDTA plasma is used given that both treatments of blood will affect the line shape of lipoproteins in the NMR spectra? Presumably they mean they can combine data from separate studies that have either used EDTA plasma or serum? This needs to be stated.

Reviewer #3 (Remarks to the Author):

(1) In general, the description on the statistical methods is minimal and not clear. For example, "adjustment for multiple testing using forward and backward stepwise selection". Forward and backward stepwise selections are methods for model selection, but not to adjust multiple testing. It is not clear how the "meta-analyses" and "survival analysis" were performed.

(2) Metabolite levels are affected by clinical and demographical variables. It is unclear whether these variables have been adjusted and if so, how are they adjusted?

(3) Out of the 12 cohorts, how heterogeneous are the metabolites profiles?

(4) For the "14 identified metabolites", has any of them been validated?

(5) Data availability: Based on "the data that support the findings of this study are available from the corresponding authors upon reasonable request", it is not clear what kind of requests are "reasonable".

(6) Figure 1: For the comparisons between the methods, are the differences in terms of AUC statistically significant?

Reply to Reviewers' comments

Reviewer #1 (Remarks to the Author):

The present study is the largest of its kind to correlate NMR-detectable plasma metabolic parameters with mortality. As such, the paper certainly deserves rapid publication in Nature Communications. Altogether, the methodology appears adequate and the data are convincing. Independent statistical review is recommended, however.

I only have some relatively minor suggestions.

In the Discussion the authors say "This resulted in [...] improved prediction accuracy (data not shown)." Since this is the crucial point of the manuscript, it would be good to demonstrate that prediction accuracy is indeed improved.

We acknowledge that it would indeed be better to show the improved prediction accuracy of the 14 mortality-associated metabolic biomarkers identified in our study in comparison to the 4 metabolic biomarkers previously reported by Fischer and colleagues. We therefore added a Table (Supplementary Table 6) containing the results of this comparison and added some accompanying text in the Results section (p. 14).

When reading the text without simultaneously analyzing Table 2 and 3, it is difficult to understand which among the metabolites or other analytes measured by NMR are positively or negatively influencing longevity. Some rewording might be helpful to improve the readability of the text.

We have made textual changes in the Results section of the manuscript (pp. 12-13) that clarify for which metabolic biomarkers a higher level is associated with decreased mortality and for which ones the opposite applies.

Reviewer #2 (Remarks to the Author):

The manuscript by Deleen and colleagues details an impressive meta-analysis of 44,168 individuals using NMR spectroscopy to profile metabolic changes in blood plasma/serum and investigate their association with all-cause mortality. This extends previous studies by Wurtz et al. and Fisher et al. that had previously examined cardiovascular risk and all-cause mortality risk in a sub-set of the cohorts used in the present study using ¹H NMR spectroscopy. The major selling point of the present study is a large increase in mortality cases across the cohorts which significantly adds to the power of their statistics, and in particular has expanded the number of 'metabolites' associated with all-cause mortality compared with their previous publication by Fisher et al. However, while they have significantly increased the number of individuals in the present study I do wonder what this analysis adds to the previous studies. While they show the metabolic markers are more predictive than some classically used parameters we don't get to understand why these supposed metabolites are more predictive or what metabolic processes they represent. I feel this is a real missed opportunity and could be tackled by examining correlations between classic risk factors and the metabolic risk factors detected by NMR spectroscopy – for example how much are the effects of the different metabolites mediated through BMI? Which of the metabolites could be combined with the classical risk factors to achieve a comparable risk prediction to the 14

metabolites? This would help us understand which metabolites brought in new predictive power compared with the classic risk factors. I also wondered how much their classic risk factor score would be improved by including some simple clinical chemistry such as blood total triglyceride.

We want to thank the Reviewer for pointing this out. To address this point we have added a Figure to the manuscript that shows the correlations between the 14 identified mortality-associated metabolic biomarkers and the conventional risk factors (Supplementary Figure 18). As indicated in this Figure, the correlations between the metabolic biomarkers and conventional risk factors are relatively low (Pearson correlation $< |0.5|$), with the exception of HDL cholesterol (high Pearson correlation with VLDL diameter) and triglycerides (high Pearson correlations with total lipids in chylomicrons and extremely large VLDL, the ratio of polyunsaturated fatty acids to total fatty acids, VLDL diameter, the concentrations of leucine, isoleucine, and GlycA). The highest Pearson correlation of BMI is with GlycA (0.41).

In addition, we performed some additional discrimination and reclassification analyses in which we compared a model with the 14 identified metabolic biomarkers to a model with the 14 identified metabolic biomarkers + conventional risk factors (excluding the ones that were also part of the 226 measured metabolic biomarkers, i.e. total and HDL cholesterol, triglycerides, and creatinine). This analysis showed that the combination of the metabolic biomarkers and conventional risk factors does not improve the risk prediction accuracy for 5- and 10- year mortality as compared to the biomarkers alone (see newly added Supplementary Table 7). Based on these results, we also made some textual changes in the Results section (p. 14).

I am also concerned about how many metabolites the authors really detect. There are not 226 independent metabolites detectable by NMR spectroscopy and in many cases this assay does not measure a metabolite as such but a chemical group found within a metabolite. However, the word metabolite is used even for quite confusing moieties such as the total lipid in chylomicrons or total lipids in extremely large VLDL. Also how can NMR work out a ratio of polyunsaturated fats to saturated fats when it can't predict what fatty acids are detected and hence the chain length of the lipids detected – certainly not what species are detected using only ¹H NMR spectroscopy in a complex mixture like blood. Presumably what they mean is that some of these resonances can be correlated with the concentrations of certain metabolites but there is a fundamental difference between say quantifying lactate where there are discrete resonances associated with only that metabolite or -CH₂- which could come from numerous fatty acids and in turn could be part of many different lipid species.

We agree with the Reviewer that the 226 measures derived from NRM profiling are not all considered metabolites. Therefore, we now use the term metabolic biomarkers throughout the manuscript. The details about the measurement and quantification of the metabolic biomarkers have been described previously (as mentioned in our Methods section (p. 7))^{1,2}.

Data availability – there are a number of repositories for metabolomic data and the data should be made available in keeping with Nature guidelines concerning open access to data.

After consulting the Editor, we have changed our data availability statement (pp. 10-11) to comply with the Nature guidelines.

How can the association of metabolites with risk of all-cause death be independent of whether serum or EDTA plasma is used given that both treatments of blood will affect the line shape of lipoproteins in the NMR spectra? Presumably they mean they can combine data from separate studies that have either used EDTA plasma or serum? This needs to be stated.

We indeed made this statement since we combined data from cohorts using either EDTA plasma or serum and wanted to point out that this had no influence on our results. We revised this part of the Discussion section (p. 15) to make this clearer.

Reviewer #3 (Remarks to the Author):

(1) In general, the description on the statistical methods is minimal and not clear. For example, “adjustment for multiple testing using forward and backward stepwise selection”. Forward and backward stepwise selections are methods for model selection, but not to adjust multiple testing. It is not clear how the “meta-analyses” and “survival analysis” were performed.

We agree that the description of the statistical analyses in the Methods section was sometimes a bit unclear. We therefore made some textual changes (pp. 9-10) to improve this section.

(2) Metabolite levels are affected by clinical and demographical variables. It is unclear whether these variables have been adjusted and if so, how are they adjusted?

As mentioned in the Methods section of the manuscript, all our models were “adjusted for age at sampling, sex and study-specific covariates”. The study-specific covariates are related to demography and relatedness. For example, the individuals from the PROSPER study have been collected in 3 different countries, while the LLS is a family-based study. We did not adjust for any clinical variables, since we wanted to identify metabolic biomarkers that are associated with mortality independent of the clinical status of the individuals (including use of medication). We made some small textual changes in the Methods section (p. 8) to better clarify this.

(3) Out of the 12 cohorts, how heterogeneous are the metabolites profiles?

There is indeed heterogeneity in the metabolic biomarker profiles between the different cohorts. However, we did not formally test this heterogeneity. Instead, to overcome the problem of heterogeneity, we performed log-transformation and scaling of the individual metabolic biomarkers to standard deviation (SD) units, separately for each cohort (as described in our Methods section (p. 8)). When the biomarkers are standardized to mean = 0 and SD = 1, they become readily combinable for epidemiological studies across cohorts. We did check for heterogeneity in the effect of the metabolic biomarkers on mortality between the different cohorts and reported this in Table 2 and Supplementary Table 3 of the manuscript (P (het) column).

(4) For the “14 identified metabolites”, has any of them been validated?

We assume that the Reviewer is referring to replication of the associations between the 14 metabolic biomarkers and mortality. If so, we have not validated any of the associations of our metabolic biomarkers with mortality since we used all cohorts with metabolomics data from Nightingale Health that were available to us to identify the metabolic biomarkers.

However, we agree that it would be good to see if our findings also replicate in independent cohorts. We hope that the studies that are currently measured with the same platform (such as the UK Biobank) will be able to replicate our findings.

However, it could also have been the case that the Reviewer is referring to validation of the measurement of the 14 metabolic biomarkers with another method. If so, we would like to refer to the previously published methodological paper about the Nightingale health platform in which several of the metabolic biomarkers were confirmed with routine clinical chemistry assays, gas chromatography, an enzymatic method, and/or mass spectrometry². In addition, the Nightingale health platform has been used in large genome-wide association studies and the metabolic biomarkers have been associated with biologically relevant genes³.

(5) Data availability: Based on “the data that support the findings of this study are available from the corresponding authors upon reasonable request”, it is not clear what kind of requests are “reasonable”.

After consulting the Editor, we have changed our data availability statement (pp. 10-11) to comply with the Nature guidelines.

(6) Figure 1: For the comparisons between the methods, are the differences in terms of AUC statistically significant?

Yes, this information is already provided in Table 4 in the column “Difference in C-statistic”.

References

1. Soininen P, Kangas AJ, Wurtz P, Suna T, Ala-Korpela M. Quantitative serum nuclear magnetic resonance metabolomics in cardiovascular epidemiology and genetics. *Circ Cardiovasc Genet* **8**, 192-206 (2015).
2. Wurtz P, Kangas AJ, Soininen P, Lawlor DA, Davey Smith G, Ala-Korpela M. Quantitative Serum Nuclear Magnetic Resonance Metabolomics in Large-Scale Epidemiology: A Primer on -Omic Technologies. *Am J Epidemiol* **186**, 1084-1096 (2017).
3. Kettunen J, *et al.* Genome-wide study for circulating metabolites identifies 62 loci and reveals novel systemic effects of LPA. *Nat Commun* **7**, 11122 (2016).

Reviewers' comments:

Reviewer #1 (Remarks to the Author):

None

Reviewer #2 (Remarks to the Author):

While I thank the authors for considering my comments I do feel they have rather missed the point of them and as a result this modified paper doesn't really address my concerns. Specifically:

Changing the name from metabolites to metabolic biomarkers doesn't really help the reader understand exactly what is measured. Without being able to relate the quantities measured by the NMR spectra back to actual metabolites it makes it difficult to hypothesize a plausible mechanism. Without a mechanism to test how can one go further than correlation to causation?

Data availability - it still has not been made available so no one can check the validity of the paper. It needs to be made available through a repository. More importantly the algorithms used need to be made available so others can test the validity of the approach. Neither of these things are available.

If none of the clinical measurements correlate with the metabolic biomarkers detected, just what are the metabolic biomarkers proxies for? Being able to hypothesize a mechanism would greatly enhance the discussion. This is particularly confusing given their comment 4 to reviewer 3 where they tell us that the metabolic biomarkers measured by NMR spectroscopy do correlate with clinical measurements or mass spectrometry measurements of certain metabolites. I agree with reviewer 3 that some validation is needed in terms of understanding what metabolites are really driving these models.

An alternative is to produce models corrected for various confounding factors as suggested by reviewer 3. This would tell us if the metabolic biomarkers are proxies for anything already measured. This should be done if we are to have any understanding of what they are measuring (and is pretty standard in most epidemiology papers where multiple factors influence a given disease process).

Reviewer #3 (Remarks to the Author):

All of my comments have been addressed and I have no further comments.

Reply to Reviewers' comments

We sincerely hope that our response below has clarified some of the comments that prompted discussion by Reviewer 2. We have made some textual additions and provided the scripts we have used for the pre-processing and analyses of our data, so one should be able to assess the validity of our approach. We are hesitant to add **Table R1** and **Figure R1** to our manuscript, since we feel that these do not add anything to the message of the manuscript and are merely created for support of our response to the comments of Reviewer 2.

Editor:

After receiving the referees' comments we discussed referee 2's comments with referee 3. Based on this we would like to highlight the referees' request to share the algorithm and data, or as much of it as possible, with the referees to allow them to assess the analysis.

It seems that Reviewer 2 is referring to the 'raw' peak data and the algorithms used to quantify the metabolic biomarkers from these peaks. However, as mentioned in our previous conversation by email, this data is not available to us. The algorithms used to quantify the metabolic phenotypes from the spectral data are Nightingale Health Ltd. intellectual property rights and the company will not publish those as the quantitation models are the core of the company's product. However, the quantified NMR measures from Nightingale Health are available through the different sources mentioned in the data availability statement. The same quantified NMR measures have been used in numerous published epidemiological studies (see <https://nightingalehealth.com/publications> for an overview), of which many in leading journals (e.g. ^{1,2,3}), all in the absence of the protected algorithms. Moreover, we have now added the most important scripts that we used for the scaling of the metabolic biomarkers, single cohort analyses (for the Alpha Omega Cohort, as example), and meta-analyses as supplementary files, so one should be able to assess the validity of our approach. We have also added some accompanying text to the data availability statement (p. 11).

Furthermore, referee 3 recommends comparing the results from an analysis with and without adjustment for clinical variables and to add results from an analysis of UK Biobank data.

With respect to the first part of the comment, we have performed an additional analysis in which we determined the effect of the metabolic biomarkers on all-cause mortality in the FINRISK 1997 cohort, with and without adjustment for the conventional risk factors (see our response to the last comment of Reviewer 2 below).

Regarding the second part of the comment, we have the impression that Reviewer 3 has misunderstood our response to the 4th of his previous comments. There is no metabolomics data from Nightingale Health available yet for UK Biobank (the public release of the data for the first 100,000 samples is scheduled in August 2020). Hence, we are unable to use this study for our analyses. However, we want to highlight here again that we included 12 cohorts in our meta-analysis and that the associations of the metabolic biomarkers with all-cause mortality are consistent across these cohorts (see Figure S2-S15), which shows the reproducibility of our findings.

Reviewer #2 (Remarks to the Author):

While I thank the authors for considering my comments I do feel they have rather missed the point of them and as a result this modified paper doesn't really address my concerns. Specifically:

Changing the name from metabolites to metabolic biomarkers doesn't really help the reader understand exactly what is measured. Without being able to relate the quantities measured by the NMR spectra back to actual metabolites it makes it difficult to hypothesize a plausible mechanism. Without a mechanism to test how can one go further than correlation to causation?

We would like to refer to the previously published methodological papers about the Nightingale Health for more details regarding the quantification of the metabolic biomarkers.^{4,5} In addition, we have created a Figure (**Figure R1**) based on public material from Nightingale Health, which has been used in several previously published manuscripts (i.e.^{1,4,6,7}) to show that the quantified NMR spectra are highly correlated with the corresponding metabolic biomarkers when measured with another method. As depicted in this Figure, several of the 14 metabolic biomarkers we identified to be associated with all-cause mortality (i.e. the concentrations of glucose, albumin, leucine, isoleucine, and valine and the ratio of polyunsaturated fatty acids to total fatty acids) have already been ‘validated’ with other techniques (i.e routine clinical chemistry assays, gas chromatography, an enzymatic method, and/or mass spectrometry). Furthermore, the metabolic biomarkers measured using the Nightingale Health platform have been used in numerous published epidemiological studies (see <https://nightingalehealth.com/publications> for an overview), of which many in leading journals (e.g.^{1,2,3}). The genetic work based on the Nightingale Health platform also underscores that the labels given to the metabolic biomarkers are correct and are associated with biologically relevant and plausible genes.^{8,9,10} We have made some textual changes in the Results section (pp. 7-8).

Data availability - it still has not been made available so no one can check the validity of the paper. It needs to be made available through a repository. More importantly the algorithms used need to be made available so others can test the validity of the approach. Neither of these things are available.

It seems that the reviewer is referring to the ‘raw’ peak data and the algorithms used to quantify the metabolic biomarkers from these peaks. However, all our samples have been measured using the Nightingale Health platform. After measuring, one receives back the quantified measures in the units described in our **Table S1**. The company only provides quantified metabolic biomarkers for which they are confident that they are reliable detectable in the NMR spectral data. The algorithms used to quantify the metabolic phenotypes from the spectral data are Nightingale Health Ltd. intellectual property rights and the company will not publish those as the quantitation models are the core of the company’s product. However, the quantified NMR measures from Nightingale Health are available through the different sources mentioned in the data availability statement. Moreover, we have now added the most important scripts that we used for the scaling of the metabolic biomarkers, single cohort analyses (for the Alpha Omega Cohort, as example), and meta-analyses as supplementary files, so one should be able to assess the validity of our approach. We have also added some accompanying text to the data availability statement (p. 11).

If none of the clinical measurements correlate with the metabolic biomarkers detected, just what are the metabolic biomarkers proxies for? Being able to hypothesize a mechanism would greatly enhance the discussion. This is particularly confusing given their comment 4 to reviewer 3 where they tell us that the metabolic biomarkers measured by NMR spectroscopy do correlate with clinical measurements or mass spectrometry measurements of certain metabolites. I agree with reviewer 3 that some validation is needed in terms of understanding what metabolites are really driving these models.

We performed our analyses to identify metabolic biomarkers that independently associate with all-cause mortality. These biomarkers were not selected based on their correlation with clinical measurements so they could potentially be used as proxies. We wanted to know if the metabolic biomarkers are better in predicting all-cause mortality than conventional risk factors that were measured in the same dataset (i.e. systolic blood pressure, body mass index, smoking status, alcohol consumption, prevalent diabetes, cardiovascular disease and cancer, and sex) and we showed that this is indeed the case. Our response to the previous comment of Reviewer 3 was just to show that the quantified metabolic biomarkers actually represent what they should (e.g. that the metabolic biomarker “triglycerides” is indeed representing triglycerides as measured with routine clinical chemistry assays). The correlation analyses we performed as a response to the 1st of the Reviewer’s previous comments were only to show that the metabolic biomarkers do not fully correlate with the conventional risk factors and likely reflect health aspects not covered by those. To be clear, we did not present this as an argument to show that the metabolic biomarkers represent conventional risk factors measured with routine clinical chemistry assays. In our opinion, it is really interesting that the status of an individual based on metabolic biomarkers alone can tell more about his/her mortality risk than the clinical measures tested in our study. We think that our results warrant further studies to determine which health conditions are further reflected by the metabolic biomarkers and by what mechanisms. Such research is exemplified by previous work on the relation between metabolic biomarkers and all-cause mortality,¹¹ which opened up new avenues for research into the GlycA metabolite.¹² We have made some textual changes in the Discussion section (p. 16).

An alternative is to produce models corrected for various confounding factors as suggested by reviewer 3. This would tell us if the metabolic biomarkers are proxies for anything already measured. This should be done if we are to have any understanding of what they are measuring (and is pretty standard in most epidemiology papers where multiple factors influence a given disease process).

To determine if the effect of the metabolic biomarkers on all-cause mortality could be explained by their correlation with conventional risk factors we performed an additional analysis in which we determined the effect of the metabolic biomarkers on all-cause mortality in the FINRISK 1997 cohort, with and without adjustment for the conventional risk factors. In **Table R1**, we show the association of (1) conventional risk factors, (2) the 14 metabolic biomarkers and (3) the combination of conventional risk factors and metabolic biomarkers with all-cause mortality (when combined into one model). When comparing the results from model 2 and 3, one could see that the association of most of the 14 metabolic biomarkers with all-cause mortality is not influenced by the addition of the conventional risk factors. However, for the metabolic biomarker Valine, the addition of the conventional risk factors decreased the association with all-cause mortality, which may be due to its involvement in diabetes (this is also observed in the reverse analysis, i.e. the association of prevalent diabetes with all-cause mortality (model 1), which is largely reduced after addition of the metabolic biomarkers

(model 3)). In addition, the effect of BMI disappears after adding the metabolic biomarkers, which is likely explained by its correlation with several of the biomarkers, as depicted in **Figure S18**. Together, these results indicate that the metabolic biomarkers contain information about the health status and mortality risk of an individual that is not captured by the conventional risk factors. For example, some of the identified metabolic biomarkers reflect obesity-related effects that are not covered by BMI, such as abdominal fat mass as measured by dual-energy X-ray absorptiometry (DEXA).¹³ As mentioned in our response to the previous comment of the Reviewer, this opens up new avenues for research to elucidate the mechanism by which these metabolic biomarkers influence mortality risk.

Table R1. Association of conventional risk factors and the 14 identified metabolic biomarkers with all-cause mortality in the FINRISK 1997 cohort. The conventional risk factors and the 14 identified metabolic biomarkers were tested separately and combined into one model.

Parameter	Conventional risk factors			Metabolic biomarkers			Conventional risk factors + metabolic biomarkers		
	HR	95% CI	P	HR	95% CI	P	HR	95% CI	P
Systolic blood pressure	1.08	1.01 - 1.14	0.015				1.06	1.00 - 1.12	0.068
Body mass index	1.09	1.02 - 1.16	0.010				0.98	0.91 - 1.05	0.516
Smoking status	2.36	2.06 - 2.71	0.00E+00				1.95	1.69 - 2.25	0.00E+00
Alcohol consumption	1.00	1.00 - 1.00	4.59E-08				1.00	1.00 - 1.00	1.11E-06
Prevalent diabetes	1.64	1.39 - 1.94	5.53E-09				1.35	1.12 - 1.63	0.002
Prevalent cardiovascular disease	1.87	1.56 - 2.25	1.95E-11				1.74	1.44 - 2.10	5.68E-09
Prevalent cancer	2.08	1.64 - 2.64	1.49E-09				1.93	1.52 - 2.45	6.25E-08
Sex	0.65	0.57 - 0.75	1.86E-10	0.53	0.46 - 0.60	0.00E+00	0.63	0.55 - 0.73	1.76E-10
GlycA				1.40	1.25 - 1.56	3.23E-09	1.34	1.20 - 1.51	3.02E-07
Alb				0.90	0.83 - 0.97	0.005	0.90	0.83 - 0.97	0.006
VLDL-D				0.97	0.86 - 1.10	0.638	0.98	0.87 - 1.11	0.776
PUFA/FA				0.74	0.69 - 0.79	0.00E+00	0.77	0.72 - 0.83	4.61E-13
Glc				1.14	1.09 - 1.19	4.91E-09	1.10	1.05 - 1.15	1.29E-04
Leu				0.78	0.64 - 0.94	0.010	0.77	0.63 - 0.93	0.007
Phe				1.23	1.12 - 1.35	2.10E-05	1.20	1.09 - 1.31	1.46E-04
XXL-VLDL-L				0.78	0.69 - 0.88	4.23E-05	0.80	0.71 - 0.90	2.44E-04
S-HDL-L				0.95	0.90 - 1.02	0.155	0.94	0.88 - 1.00	0.055
AcAce				1.06	1.00 - 1.13	0.055	1.04	0.98 - 1.11	0.180
Ile				1.16	0.98 - 1.37	0.083	1.19	1.01 - 1.41	0.038
His				0.92	0.86 - 0.99	0.024	0.92	0.86 - 0.99	0.030
Val				0.89	0.78 - 1.01	0.077	0.94	0.82 - 1.07	0.358
Lac				1.06	1.00 - 1.14	0.064	1.04	0.97 - 1.11	0.254

Figure R1. Correlations between metabolic biomarkers measured with routine clinical chemistry assays, gas chromatography, an enzymatic method, and/or mass spectrometry and NMR (based on data from ^{1, 4, 6, 7}).

References

1. Wurtz P, *et al.* Metabolite profiling and cardiovascular event risk: a prospective study of 3 population-based cohorts. *Circulation* **131**, 774-785 (2015).
2. Auro K, *et al.* A metabolic view on menopause and ageing. *Nat Commun* **5**, 4708 (2014).
3. Kujala UM, *et al.* Long-term leisure-time physical activity and serum metabolome. *Circulation* **127**, 340-348 (2013).
4. Wurtz P, Kangas AJ, Soininen P, Lawlor DA, Davey Smith G, Ala-Korpela M. Quantitative Serum Nuclear Magnetic Resonance Metabolomics in Large-Scale Epidemiology: A Primer on -Omic Technologies. *Am J Epidemiol* **186**, 1084-1096 (2017).
5. Soininen P, *et al.* High-throughput serum NMR metabolomics for cost-effective holistic studies on systemic metabolism. *Analyst* **134**, 1781-1785 (2009).
6. Tynkkynen J, *et al.* Association of branched-chain amino acids and other circulating metabolites with risk of incident dementia and Alzheimer's disease: A prospective study in eight cohorts. *Alzheimers Dement* **14**, 723-733 (2018).
7. Holmes MV, *et al.* Lipids, Lipoproteins, and Metabolites and Risk of Myocardial Infarction and Stroke. *J Am Coll Cardiol* **71**, 620-632 (2018).
8. Tukiainen T, *et al.* Detailed metabolic and genetic characterization reveals new associations for 30 known lipid loci. *Hum Mol Genet* **21**, 1444-1455 (2012).
9. Kettunen J, *et al.* Genome-wide association study identifies multiple loci influencing human serum metabolite levels. *Nat Genet* **44**, 269-276 (2012).
10. Kettunen J, *et al.* Genome-wide study for circulating metabolites identifies 62 loci and reveals novel systemic effects of LPA. *Nat Commun* **7**, 11122 (2016).
11. Fischer K, *et al.* Biomarker profiling by nuclear magnetic resonance spectroscopy for the prediction of all-cause mortality: an observational study of 17,345 persons. *PLoS Med* **11**, e1001606 (2014).
12. Ritchie SC, *et al.* The Biomarker GlycA Is Associated with Chronic Inflammation and Predicts Long-Term Risk of Severe Infection. *Cell Syst* **1**, 293-301 (2015).

13. Bogl LH, *et al.* Abdominal obesity and circulating metabolites: A twin study approach. *Metabolism* **65**, 111-121 (2016).

Reviewers' comments:

Reviewer #2 (Remarks to the Author):

Many thanks for the authors for responding to my comments. I am afraid these responses have raised more concerns for me. The quantification of the metabolites reported in the paper relies on NMR spectroscopy which detects a number of resonances associated with each metabolite. My concern is that the authors only receive a "peak list" and cannot as a result really vouch for the validity of the approach or the analyses performed by the company. As a reviewer I need to be convinced that the analyses reported in a paper are correct and by being one removed from those that have analysed the data I cannot see how I can be convinced. I am aware that this particular approach has been used previously by others but equally in none of those publications have the issues I raise been addressed. Furthermore, these are not unusual requests for metabolomic datasets performed by ¹H NMR spectroscopy. I still cannot see why a company should have special treatment compared with academic scientists in terms of what they report. I have addressed my specific concerns below which I hope illustrates my concerns.

When I requested the raw data I am requesting the NMR spectroscopy FIDs (or spectra). This is not subject to any patent by Nightingale Health Ltd and will have been acquired on a NMR spectrometer developed by a third party (most likely Bruker). Such spectra cannot be traced back to an individual so should not break volunteer confidentiality. While the access to the raw data would not allow people to re-create the algorithm used for peak fitting it would at least allow people to create their own analyses on the dataset and to reproduce some of the analyses subsequently performed in the paper. This needs to be made available in terms of the reproducibility of science.

If the peak fitting algorithm cannot be reported for financial reasons then some screen shots of how it does in terms of fitting some of the key metabolites/resonances would be useful. Are their residual signals below these peaks? What co-resonant features are there? Normally, in a paper using NMR spectroscopy the authors would quote a chemical shift(s) and coupling pattern. This is not done in the paper at all. While I appreciate the figure concerning cross validation of the results this is not performed on this particular dataset and we can also see that it performs less well for some of the metabolites they discuss in the paper: leucine seems poorly modelled and one wonders whether this is down to co-resonant metabolites, the correlation for glucose seems to be driven by points at high glucose, the correlation for creatinine has a lot less points compared with the others. What resonances are used for MUFA, PUFA, omega-6 and DHA? Are they all correlated and not really independent variables in this dataset? I ask as ¹H NMR relies on the detection of CH=CH and CH₂-CH=CH, so how confident can one be in quantifying all these metabolites with so few resonances? Hopefully these comments can show my concerns of over interpretation of the data.

The authors state: "We think that our results warrant further studies to determine which health conditions are further reflected by the metabolic biomarkers and by what mechanisms. Such research is exemplified by previous work on the relation between metabolic biomarkers and all-cause mortality, which opened up new avenues for research into the GlycA metabolite." I still think the mechanism should be investigated further. My concern is that without knowing what was measured how can we investigate the mechanism? We need more information if we are to have more insight into why all-cause mortality is potentially being predicted.

Reviewer #3 (Remarks to the Author):

Given the variations among the 12 cohorts, it's hard to justify the use of fixed effect model to test the heterogeneity, and some of the test results are significant.

Reviewer #2 (Remarks to the Author):

Many thanks for the authors for responding to my comments. I am afraid these responses have raised more concerns for me. The quantification of the metabolites reported in the paper relies on NMR spectroscopy which detects a number of resonances associated with each metabolite. My concern is that the authors only receive a “peak list” and cannot as a result really vouch for the validity of the approach or the analyses performed by the company. As a reviewer I need to be convinced that the analyses reported in a paper are correct and by being one removed from those that have analysed the data I cannot see how I can be convinced. I am aware that this particular approach has been used previously by others but equally in none of those publications have the issues I raise been addressed. Furthermore, these are not unusual requests for metabolomic datasets performed by ^1H NMR spectroscopy. I still cannot see why a company should have special treatment compared with academic scientists in terms of what they report. I have addressed my specific concerns below which I hope illustrates my concerns.

When I requested the raw data I am requesting the NMR spectroscopy FIDs (or spectra). This is not subject to any patent by Nightingale Health Ltd and will have been acquired on a NMR spectrometer developed by a third party (most likely Bruker). Such spectra cannot be traced back to an individual so should not break volunteer confidentiality. While the access to the raw data would not allow people to re-create the algorithm used for peak fitting it would at least allow people to create their own analyses on the dataset and to reproduce some of the analyses subsequently performed in the paper. This needs to be made available in terms of the reproducibility of science. If the peak fitting algorithm cannot be reported for financial reasons then some screen shots of how it does in terms of fitting some of the key metabolites/resonances would be useful. Are their residual signals below these peaks? What co-resonant features are there? Normally, in a paper using NMR spectroscopy the authors would quote a chemical shift(s) and coupling pattern. This is not done in the paper at all.

As mentioned in our previous responses, we are not able to share the NMR spectroscopy FIDs from the Nightingale Health platform nor the algorithms used for the quantification of the metabolic biomarkers, since this is part of the agreement with Nightingale Health. However, we have NMR spectroscopy FIDs available for close to 3,000 samples from one of our studies (Leiden Longevity Study). These samples were measured some years ago with in-house NMR by Dr. Aswin Verhoeven and later with the Nightingale Health platform as part of a larger effort.¹ The correlations between the metabolic biomarkers as measured on the two platforms is high ($r > 0.8$ for all mortality-associated metabolic biomarkers for which we have overlapping quantified metabolic biomarker data from both platforms, see **Supplementary Figure 19**). We would thus like to suggest to the Reviewer to take a look at some of these spectra and the accompanying workflow used to quantify them (which we have attached as supplementary data) to see if he/she thinks this is trustable. To access the workflow, the Reviewer should install the KNIME Analytics Platform (<https://www.knime.com/knime-software/knime-analytics-platform>) and subsequently import the “KIMBLE LLS partial.knwf” file via the option ‘Import KNIME workflow...’ under ‘File’. We have also imported the metabolic biomarker values from the same individuals as quantified with the Nightingale Health platform, so the Reviewer can see that these are very consistent, especially for the metabolic biomarkers associated with mortality (see output of ‘Column to Grid’ in the workflow (accessible via a right mouse click and subsequent selection of the ‘Grid table’ at the bottom)). More details on KIMBLE can be found here: <https://cpm.lumc.nl/kimble/>. Due

to the inclusion of the in-house NMR data from Leiden, we have added Dr. Verhoeven as a co-author to the manuscript. As described in more detail below, we have also provided some additional data to show the consistency of the metabolic biomarkers across multiple commercially available platforms (**Supplementary Table 8**). These novel data generally show high analytical consistency with other non-NMR based metabolomics platforms, providing further evidence that the metabolic biomarkers highlighted in this manuscript are reflecting the labels assigned. In addition, we have added a Figure from one of the previous published papers using the Nightingale Health platform (**Figure R1**),² which shows an example of spectral data identifying all the resonances representing our 14 identified metabolic biomarkers associated with mortality in individuals before and after treatment with statins. This Figure also shows the resonance used for quantifying the different fatty acid-related metabolic biomarkers mentioned below by the Reviewer. The chemical shifts and J-couplings for individual metabolic biomarker signals are similar to those that are widely quoted in the spectroscopy literature (e.g. ³, as described in the methodological papers regarding the Nightingale Health platform).^{4,5} Last but not least, we have added Peter Würtz, the Scientific Director of Nightingale Health, as a co-author to the manuscript so he can act a guarantor for the integrity of the metabolic biomarker quantifications performed by Nightingale Health. We hope that we have thus provided enough data and information for the Reviewer to check the validity of our approach.

We have added a new Supplementary Figure (Supplementary Figure S19) and Table (Supplementary Table 8) and we have made some textual changes in the Results section (pp. 14-15) of the manuscript.

While I appreciate the figure concerning cross validation of the results this is not performed on this particular dataset and we can also see that it performs less well for some of the metabolites they discuss in the paper: leucine seems poorly modelled and one wonders whether this is down to co-resonant metabolites, the correlation for glucose seems to be driven by points at high glucose, the correlation for creatinine has a lot less points compared with the others.

To accommodate the Reviewer's point, we have now additionally added a Table including the correlations of 7 out of the 14 identified metabolic biomarkers associated with mortality as measured with different metabolomics platforms (i.e. Metabolon and/or Biocrates) (**Supplementary Table 8**). This Table shows that most of our identified metabolites could be validated very well when using another metabolomics platform. For GlycA, we would like to refer to one of our previous publications in which we calculated the correlations with the main proteins underlying the signal.⁶ Given that this is a composite signal, which also contains a lipid signal (correlating with the total lipids in chylomicrons and extremely large VLDL in our study), the correlations are lower than for the other metabolic biomarkers. We were not able to perform cross-platform correlations of the lipoprotein subclasses, since there is no published data, neither a clear gold standard, available for these metabolic biomarkers. Given that we have already showed cross validation results for the ratio of polyunsaturated fatty acids to total fatty acids ($r = 0.94$) and albumin levels ($r = 0.80$) in our previous response, we have now provided cross validation (with another technique or platform) for 11 out of the 14 identified metabolic biomarkers associated with mortality. Together with the NMR spectroscopy FIDs and accompanying workflow provided for the in-house NMR, showing very strong consistency with the results from the Nightingale Health platform, we hope that this is sufficient to convince the Reviewer that the labels for our metabolic biomarkers provided by Nightingale Health are valid. While we acknowledge that the cross validations

that we provided in our previous response were not made in the same datasets that are assessed here, we would like to emphasize that the same NMR-based method for metabolite quantification is used across all of the cohorts included in our meta-analysis and that it would be infeasible to conduct the consistency testing of the metabolic biomarkers in all the 12 used cohorts (n = 44,168).

We also want to point out that the data for creatinine we used in our previous response, showing a very high correlation ($r = 0.98$) was based on data from 4,635 samples⁷, so the number of points included in this Figure is substantial. However, we acknowledge that this may not have been clearly visible in the Figure. Moreover, given the large number of samples used to calculate the correlation for glucose ($r = 0.96$, $n = 2,749$)⁸ and the small subset of samples with high glucose, it is very unlikely that the correlation is driven by this small subset. In addition, the previously mentioned correlations of the branched-chain amino acids, which were based on comparison of the Nightingale Health-derived metabolic biomarkers with Biocrates p180-derived values for the same biomarkers in the FINRISK 1997 cohort, was indeed relatively low, but, as shown in **Supplementary Table 8**, these metabolic biomarkers show much higher cross-platform correlations in other studies.

What resonances are used for MUFA, PUFA, omega-6 and DHA? Are they all correlated and not really independent variables in this dataset? I ask as 1H NMR relies on the detection of CH=CH and CH2-CH=CH, so how confident can one be in quantifying all these metabolites with so few resonances? Hopefully these comments can show my concerns of over interpretation of the data.

The resonances used to quantify the fatty acid-related metabolic biomarkers are shown in **Figure R1**. Moreover, we have added a correlation plot of all metabolic biomarkers to convince the Reviewer that the fatty acid-related metabolic biomarkers are highly correlated (**Figure R2**). We again would like to emphasize that the correlations for the fatty acid-related metabolic biomarkers measured using the Nightingale Health platform and gas chromatography is consistently high ($r > 0.9$, see **Figure R1** from our previous response), despite the analytical consistency was assessed years apart, so we feel confident that the fatty acid-related metabolic biomarkers have been reliably quantified.

The authors state: “We think that our results warrant further studies to determine which health conditions are further reflected by the metabolic biomarkers and by what mechanisms. Such research is exemplified by previous work on the relation between metabolic biomarkers and all-cause mortality, which opened up new avenues for research into the GlycA metabolite.” I still think the mechanism should be investigated further. My concern is that without knowing what was measured how can we investigate the mechanism? We need more information if we are to have more insight into why all-cause mortality is potentially being predicted.

We appreciate the Reviewer's point to improve the traceability of what is measured, but we respectfully disagree that we do not know what was measured. We hope that the provided Figures and Table justify the labelling of the metabolic biomarkers, such as the amino acids, by the individual metabolite name (rather than some chemical shift). We believe this can actually help to study the etiological underpinnings of the reported associations regardless of the analytical assay used for the quantification. Moreover, the aim of this epidemiological study was to identify metabolic biomarkers associated with mortality (i.e. biomarker identification). Hence, it was not meant to study the mechanisms underlying the reported associations, which is a huge effort on itself and could be done in follow-up studies.

Reviewer #3 (Remarks to the Author):

Given the variations among the 12 cohorts, it's hard to justify the use of fixed effect model to test the heterogeneity, and some of the test results are significant.

The aim of our study was to identify metabolic biomarkers associated with mortality that are showing these effects independent of the background of the individuals. Hence, the best way to select such metabolic biomarkers is by using a fixed-effect model, which assumes equal effects in all studied populations. We agree with the Reviewer that some of the P-values for heterogeneity are indeed nominal significant ($P(\text{het}) < 0.05$). However, after adjustment for multiple testing, only PUFA/FA remains significant ($P(\text{het}) = 8.65 \times 10^{-5}$). To accommodate the Reviewer, we have added a Table (**Table R1**), which shows that the effects from a random-effect model are very similar, but due to the loss of power the significance for some metabolic biomarkers is slightly reduced. In particular, PUFA/FA remains one of the strongest metabolic biomarkers if evaluated with a random-effects model. Hence, we feel that the use of a fixed-effect model is justified. In addition, the assessment of the improved prediction in mortality (by calculation of the weights in the Estonian Biobank cohort and subsequent testing in the FINRISK 1997 cohort) is not affected by this issue. If there would be substantial heterogeneity between the metabolic biomarker relations in these large cohorts, no added predictive value would have been observed.

Table R1. Comparison of results using a fixed- and random-effects model for the 14 identified metabolic biomarkers associated with all-cause mortality.

Biomarker	Full name	Fixed-effect			Random-effect			I ²	P (het)
		HR	95% CI	P	HR	95% CI	P		
XXL-VLDL-L	Total lipids in chylomicrons and extremely large VLDL	0.80	0.75 - 0.85	1.53 x 10 ⁻¹³	0.80	0.75 - 0.86	2.87 x 10 ⁻¹¹	0.08	0.363
S-HDL-L	Total lipids in small HDL	0.87	0.84 - 0.90	5.98 x 10 ⁻¹⁹	0.85	0.80 - 0.90	4.62 x 10 ⁻⁹	0.52	0.018
VLDL-D	Mean diameter for VLDL particles	0.85	0.80 - 0.90	8.51 x 10 ⁻⁸	0.85	0.79 - 0.91	8.62 x 10 ⁻⁶	0.21	0.241
PUFA/FA	Ratio of polyunsaturated fatty acids to total fatty acids (%)	0.78	0.75 - 0.80	1.06 x 10 ⁻⁴⁷	0.77	0.71 - 0.83	5.71 x 10 ⁻¹²	0.71	8.65 x 10 ⁻⁵
Glc	Glucose	1.16	1.13 - 1.19	2.22 x 10 ⁻²⁹	1.16	1.11 - 1.22	6.48 x 10 ⁻¹⁰	0.56	0.008
Lac	Lactate	1.06	1.03 - 1.10	6.24 x 10 ⁻⁵	1.07	1.03 - 1.11	0.001	0.28	0.173
His	Histidine	0.93	0.90 - 0.96	1.15 x 10 ⁻⁵	0.93	0.90 - 0.97	3.33 x 10 ⁻⁴	0.24	0.213
Ile	Isoleucine	1.23	1.14 - 1.32	2.14 x 10 ⁻⁸	1.19	1.07 - 1.33	0.001	0.39	0.078
Leu	Leucine	0.82	0.76 - 0.89	7.34 x 10 ⁻⁷	0.84	0.76 - 0.94	0.003	0.35	0.109
Val	Valine	0.87	0.82 - 0.92	1.04 x 10 ⁻⁶	0.86	0.81 - 0.92	3.99 x 10 ⁻⁶	0.07	0.376
Phe	Phenylalanine	1.13	1.09 - 1.17	2.39 x 10 ⁻¹²	1.15	1.09 - 1.21	9.11 x 10 ⁻⁷	0.44	0.052
AcAce	Acetoacetate	1.08	1.05 - 1.11	1.73 x 10 ⁻⁸	1.08	1.04 - 1.12	7.40 x 10 ⁻⁵	0.35	0.108
Alb	Albumin	0.89	0.87 - 0.92	9.96 x 10 ⁻¹³	0.89	0.84 - 0.94	9.10 x 10 ⁻⁶	0.52	0.017
GlycA	Glycoprotein acetyls	1.32	1.27 - 1.38	7.45 x 10 ⁻⁴¹	1.34	1.26 - 1.43	9.49 x 10 ⁻¹⁹	0.45	0.046

Figure R1. Representative NMR spectral data of individuals before and after statin use illustrating the approximate molecular content as well as signal assignments (adapted from ²).

Figure R2.

[REDACTED]

References

1. Verhoeven A, Slagboom E, Wuhrer M, Giera M, Mayboroda OA. Automated quantification of metabolites in blood-derived samples by NMR. *Anal Chim Acta* **976**, 52-62 (2017).
2. Wurtz P, *et al.* Metabolomic Profiling of Statin Use and Genetic Inhibition of HMG-CoA Reductase. *J Am Coll Cardiol* **67**, 1200-1210 (2016).
3. Nicholson JK, Foxall PJ, Spraul M, Farrant RD, Lindon JC. 750 MHz ¹H and ¹H-¹³C NMR spectroscopy of human blood plasma. *Anal Chem* **67**, 793-811 (1995).
4. Soininen P, *et al.* High-throughput serum NMR metabonomics for cost-effective holistic studies on systemic metabolism. *Analyst* **134**, 1781-1785 (2009).
5. Soininen P, Kangas AJ, Wurtz P, Suna T, Ala-Korpela M. Quantitative serum nuclear magnetic resonance metabolomics in cardiovascular epidemiology and genetics. *Circ Cardiovasc Genet* **8**, 192-206 (2015).
6. Ritchie SC, *et al.* The Biomarker GlycA Is Associated with Chronic Inflammation and Predicts Long-Term Risk of Severe Infection. *Cell Syst* **1**, 293-301 (2015).
7. Holmes MV, *et al.* Lipids, Lipoproteins, and Metabolites and Risk of Myocardial Infarction and Stroke. *J Am Coll Cardiol* **71**, 620-632 (2018).
8. Wurtz P, Kangas AJ, Soininen P, Lawlor DA, Davey Smith G, Ala-Korpela M. Quantitative Serum Nuclear Magnetic Resonance Metabolomics in Large-Scale Epidemiology: A Primer on -Omic Technologies. *Am J Epidemiol* **186**, 1084-1096 (2017).
9. Wurtz P, *et al.* Metabolite profiling and cardiovascular event risk: a prospective study of 3 population-based cohorts. *Circulation* **131**, 774-785 (2015).

REVIEWERS' COMMENTS:

Reviewer #2 (Remarks to the Author):

I thank the authors for their responses. My major concern with the previous version of the manuscript was a lack of data availability so a reviewer or reader could judge the validity of the approach. By making available the primary data for nearly 3000 NMR spectra from the Leiden Longevity Study is welcome - as is the previously developed KNIME workflow. However, their suggestion to make this available through Dropbox is not that useful. I have tried to access the data and cannot without the authors' permission to access the data. While I am happy to give up my anonymity for the sake of this review this solution still only makes the data available at the whim of the authors, and its also not a long term solution. The data should be made available through an open resource such as MetaboLights. In this manner the data will always be available. It also needs to be made available with the comparable results from the Nightingale software so the reader can check that the result is indeed consistent. This would go a long way to ensuring the study is reproducible.

If the authors can address this one point I am happy to ignore my previous request for a mechanistic justification of the metabolic features they have identified to be associated with all cause mortality - it would still be useful to understand the underlying mechanism, even for a biomarker(!), as it would give us confidence in the results.

Reviewer #3 (Remarks to the Author):

"To deal with the correlation between biomarkers, a model with all independent biomarkers was constructed after adjusting for multiple testing using forward and backward selection."
It is not clear how this was done, especially how the forward and backward selection was performed to adjust multiple testing.

Reviewer #2 (Remarks to the Author):

I thank the authors for their responses. My major concern with the previous version of the manuscript was a lack of data availability so a reviewer or reader could judge the validity of the approach. By making available the primary data for nearly 3000 NMR spectra from the Leiden Longevity Study is welcome - as is the previously developed KNIME workflow. However, their suggestion to make this available through Dropbox is not that useful. I have tried to access the data and cannot without the authors' permission to access the data. While I am happy to give up my anonymity for the sake of this review this solution still only makes the data available at the whim of the authors, and its also not a long term solution. The data should be made available through an open resource such as MetaboLights. In this manner the data will always be available. It also needs to be made available with the comparable results from the Nightingale software so the reader can check that the result is indeed consistent. This would go a long way to ensuring the study is reproducible. If the authors can address this one point I am happy to ignore my previous request for a mechanistic justification of the metabolic features they have identified to be associated with all cause mortality - it would still be useful to understand the underlying mechanism, even for a biomarker(!), as it would give us confidence in the results.

To accommodate the Reviewer's point, the in-house NMR data from Leiden have been deposited in MetaboLights under accession code MTBLS974 (<https://www.ebi.ac.uk/metabolights/MTBLS974>). The accompanying quantified metabolic biomarker datasets from Nightingale Health are available upon request through <http://www.bbmri.nl/omics-metabolomics/>.

Reviewer #3 (Remarks to the Author):

“To deal with the correlation between biomarkers, a model with all independent biomarkers was constructed after adjusting for multiple testing using forward and backward selection.”

It is not clear how this was done, especially how the forward and backward selection was performed to adjust multiple testing.

After consulting the Editor, we have made some textual changes in the Results section (p. 9) and Methods section (p. 19) of the manuscript to better explain how the stepwise procedure to select independent metabolic biomarkers associated with mortality was performed.